# Examining Place-Based Neighborhood Factors in a Multisite Peer-Led Healthy Lifestyle Effectiveness Trial for People with Serious Mental Illness

**DOI:** 10.3390/ijerph20095679

**Published:** 2023-04-28

**Authors:** Deborah Salvo, Eugen Resendiz, Ana Stefancic, Leopoldo J. Cabassa

**Affiliations:** 1Department of Kinesiology and Health Education, The University of Texas at Austin, Austin, TX 78712, USA; 2Prevention Research Center in St. Louis, Brown School, Washington University in St. Louis, St. Louis, MO 63130, USA; 3Department of Psychiatry, Columbia University, New York, NY 10032, USA; 4Center for Mental Health Services Research, Brown School, Washington University in St. Louis, St. Louis, MO 63130, USA; ljcabassa@wustl.edu

**Keywords:** serious mental illness, obesity, healthy lifestyle interventions, physical activity, geographic information systems, built environment, social environment

## Abstract

People with severe mental illness (SMI) experience significantly higher obesity-related comorbidities and premature mortality rates than healthy populations. The physical and social characteristics of neighborhoods where people with SMI reside can play an important role in promoting or hindering healthy eating and physical activity. However, this is seldom considered when designing and testing health behavior interventions for these populations. This study used baseline data from an obesity control trial for low-income, minority people with SMI to demonstrate the utility of assessing neighborhood- and city-level place-based factors within the context of lifestyle interventions. GIS was used to create a zip-code-level social and built environment geodatabase in New York City and Philadelphia, where the trial occurred. Chi-square and t-tests were used to assess differences in the spatial distribution of health-related built and social environment characteristics between and within cities and diet and physical activity outcomes. All types of neighborhood characteristics showed significant environmental differences between and within cities. Several neighborhood characteristics were associated with participants’ baseline healthy eating and physical activity behaviors, emphasizing that place-based factors may moderate lifestyle interventions for SMI patients. Future behavioral interventions targeting place-dependent behaviors should be powered and designed to assess potential moderation by place-based factors.

## 1. Introduction

Blacks and Hispanics with serious mental illness (SMI, e.g., schizophrenia, bipolar disorder) face multiple health inequities driven by the intersection of their historically marginalized statuses in the US. Compared to the general population, people with SMI die approximately 10–25 years earlier, largely due to elevated rates of obesity, diabetes mellitus, cancer, and cardiovascular disease (CVD) [1]. Belonging to a minoritized racial/ethnic group in the US further exacerbates these health inequities in people with SMI. Compared to non-Hispanic whites with SMI, Blacks and Hispanics with SMI have an elevated risk of obesity and diabetes mellitus and face numerous barriers to accessing and receiving high-quality medical care [2,3,4]. Understanding how social determinants of health, particularly place-based neighborhood factors (e.g., food environment), impact Blacks and Hispanics with SMI is critical for eliminating these deadly health inequities.

A constellation of factors contributes to the excess morbidity and mortality associated with CVD, cancer, obesity, and diabetes mellitus in people with SMI. These include the interplay of unhealthy behaviors (e.g., higher rates of smoking and lack of physical activity), the cardiometabolic side effects of psychiatric medications (e.g., weight gain), and the lack of access to and underutilization of high-quality medical care [5]. However, these factors do not happen in a vacuum. A person’s physical health is connected and influenced by the environmental factors that surround them [6]. The choices that people with SMI make regarding their dietary habits, physical activity, whether they smoke or use substances and when, how, and where they seek medical care are influenced by many aspects of their life, including the places where they live.

The socio-ecological model of health behaviors stipulates that individual-level health behaviors, such as dietary choices and practices, physical activity, and sedentary behaviors, are influenced by a variety of factors across multiple levels, including, broadly, the individual, inter-personal, organizational, and environmental levels [7]. Socio-ecological models also suggest that these levels interact with each other. For example, the effect of neighborhood availability to recreational facilities, such as YMCA’s (an organizational-level variable), on physical activity, may be moderated by the availability of public transit and road-network connectivity (environmental-level variables) to access these types of community assets [8,9]. The individual (e.g., biological, sociodemographic, attitudes and beliefs) and inter-personal (e.g., family structure and support) determinants of healthy living among the general population and people with SMI have been extensively documented [10,11]. However, less emphasis has been placed on characterizing the environmental factors influencing the ability of people with SMI, particularly those from racial and ethnic minoritized groups in the US, to lead active and healthy lifestyles. That is, the role of *place,* defined as the social, economic, cultural, physical, natural, informational, and political context in which people reside, work and play, remains mostly unexplored for people living with SMI [12,13]. Adopting a place-based approach is critical for understanding and eliminating health inequities for people with SMI, as inter-personal strategies to maintain and enhance health, such as the ones used in healthy lifestyle interventions (e.g., healthy eating, which requires access to affordable nutritious foods), may have limited effectiveness, maintenance, and scale-up potential if the underlying environment where people live, work and play is unsupportive. This is because key obesity-related behaviors, such as healthy eating and physical activity participation, are place-dependent. As such, in this work, we use the socio-ecological model as our key guiding framework.

In the US and globally, there is a dearth of knowledge with regard to this topic, particularly among minoritized racial and ethnic groups living with SMI [14]. Moreover, few studies to date have characterized critical place-based factors that can shape the health of these historically marginalized populations, including whether and how these factors are associated with dietary behaviors and physical activity: two critical health behaviors linked to obesity and CVD risk in people with SMI and key targets of healthy lifestyle interventions [15,16]. For example, Byrne and colleagues [15] used a variety of place-based measures to examine whether the physical and social characteristics of neighborhoods in Philadelphia where people with SMI live differ from neighborhoods representative of the city’s general population. They found that the neighborhoods in which adults with SMI resided had higher levels of physical and structural inadequacy, drug-related activity, and crime when compared to neighborhoods representative of Philadelphia’s general population. Furthermore, higher levels of physical and structural inadequacy, crime, drug-related activity, social instability, and social isolation were associated with a higher concentration of persons with SMI [15].

Moreover, Young-Wolff and colleagues [16] examined the density and proximity of tobacco retailers and whether retail availability was associated with nicotine dependence in a racially and ethnically diverse sample of smokers with SMI living in San Francisco, California. They found that compared to the average San Francisco Bay Area resident, adult smokers with SMI were living in neighborhoods with two-fold greater tobacco retailer densities, and that this concertation of tobacco retailers was significantly associated with greater nicotine dependence and lower readiness to quit smoking [16]. These findings highlight how place-based factors shape the health and wellbeing of people with SMI.

Studying place-based factors can have multiple benefits for improving the health of people with SMI. First, it could help identify a new set of modifiable environmental targets for interventions and policies to reduce health inequities and premature mortality. Second, place-based factors can act as potential moderators of healthy lifestyle interventions for people with SMI, thus showingwhere these interventions work best and how to tailor them for diverse populations [17]. Lastly, a deeper understanding of the relationship between place-based factors and the health of people with SMI can be used by clinicians and organizations to help mitigate the negative impact of these environmental factors on the lives and health of people with SMI.

In the present study, we used baseline data from a trial testing the effectiveness of a group-based peer-led healthy lifestyle intervention in adults with SMI who were overweight or obese and living in supportive housing in New York City and Philadelphia [18]. People with SMI who were enrolled in this study were mostly from minoritized racial/ethnic minoritized groups, particularly Blacks and Hispanics [19]. The aims of this paper are to (1) determine if the social, urban design, physical activity, food, legal substances, and healthcare environments of the neighborhoods in which the participants enrolled in this trial lived in New York City and Philadelphia, respectively, differed significantly from those of other neighborhoods within the same city; (2) determine if the distribution of these environmental factors differed significantly across the two studied cities; and (3) explore potential differences in participants’ baseline dietary and physical-activity behaviors using place-based factors. The geospatial and statistical analyses presented in this paper are intended to serve as an illustrative case study of how place-based factors can be examined within the context of an effectiveness trial of people with SMI living in the community to gain a deepener understanding of environmental factors that shape the lives of these individuals and how they can influence important health behaviors (e.g., diet and physical activity).

## 2. Materials and Methods

### 2.1. Parent Study

Data from this study were obtained from a hybrid type-1 trial that tested the effectiveness and examined the implementation of a peer-led healthy lifestyle program (PGLB) for overweight/obese participants (defined in the parent study as having BMI > 25) with SMI in three supportive housing agencies, one in New York City and two in Philadelphia. Most participants in this trial were enrolled in the Supplementary Nutrition Assistance Program (91%) and receiving Supplemental Security Income or Social Security Disability Insurance (78%) [19]. The trial protocol and main outcomes were published elsewhere [18,19]. In brief, participants were recruited between June 2015 and January 2018. A total of 314 participants were enrolled in this trial; 190 were enrolled in the Philadelphia sites, and 124 were enrolled in the New York City site. Participants were screened for study eligibility by the study team at each agency. Independent assessors employed by the study team and not blinded to participants’ group assignment conducted face-to-face interviews at the participant’s supportive housing agency at baseline and 6-, 12-, and 18-months post-randomization. Measurement protocols are described elsewhere [18]. After the baseline interview, randomization to PGLB or usual care was conducted at the participant level in blocks of four, stratified by site. PGLB is a 12-month manualized healthy lifestyle intervention delivered by peer specialists—people with lived experience with SMI. This intervention was adapted from the Group Lifestyle Balance program to meet the need of people with SMI living in supportive housing [20]. PGLB consisted of weekly core sessions for three months, twice monthly sessions for three months, and monthly maintenance sessions for six months for a total of 22 sessions.

### 2.2. Study Design and Geospatial Data Sources

The present analysis corresponds to a cross-sectional, ecological study, integrating neighborhood-level data from secondary sources with individual-level PGLB study data collected at baseline. For the purposes of this analysis, the geospatial unit for characterizing the multiple levels of the environment of neighborhoods was the zip code (US postal code). This was decided since the parent study collected zip code data from all participants. As such, describing neighborhood characteristics occurring within zip code boundaries across the cities of New York and Philadelphia enabled us to link these ecological, zip-code-level data to the PGLB study participant baseline assessment data. Zip codes that were not inhabited were excluded from this analysis. This included zip codes for which the entirety of the area covered was a public facility (e.g., an airport) and zip codes corresponding to Postal Office Box facilities. Zip-code-level data were sourced from a range of public and proprietary geospatial data sources, including: the US Census (American Community Survey), the City of New York Open Data Portal, the City of Philadelphia Open Data Portal, the National Neighborhood Data Archive, the City of Philadelphia Metadata Catalogue, the New York City Department of City Planning data portal, the New York City Metropolitan Transportation Authority, the Southeastern Pennsylvania Transportation Authority, the YMCA data portal, the Uniform Data System Mapper, the ESRI ArcGIS Business Analysist database, and the National Neighborhood Data Archive (NaNDA).

### 2.3. Neighborhood-Level Measures

Using geographic information systems (GIS) processing software, we constructed a series of neighborhood-level indicator variables to characterize zip codes’ social and built environment [21,22,23,24] across the two cities where the PGLB study took place (New York City, NY, USA and Philadelphia, PA, USA). Variables were selected based on the types of neighborhood-level features that were found to be associated with obesity-related behaviors (physical activity and dietary behaviors) and mental health in general population studies [25,26,27,28,29,30,31].

This approach allowed us to compare the characteristics of neighborhoods where PGLB study participants resided at the time of study enrollment versus other neighborhoods in their own city, as well as overall differences between the two study cities. Both comparisons are important. On the one hand, determining if potential neighborhood-level moderators vary when comparing the areas in which participants live to the rest of their city helps to assess whether the findings might be generalizable to residents in other parts of the city, or if adaptations are necessary to translate the intervention to other settings within the city. Determining, at the individual city level, place-based differences among neighborhoods where people with SMI reside in comparison to the rest of their city is also helpful for uncovering place-based disparities affecting this group. On the other hand, some intervention studies include sites distributed across multiple cities, such as the PGLB study [19]. Cities are humankind’s most enduring and stable mode of social organization and are the dominant demographic and economic clusters in society. In urban settings, people’s everyday interactions with their environment are not limited to their neighborhoods but extend to their entire city. As such, the full range of available environmental assets, structures, services, design, and social elements existing in a city at large can influence daily living, health, and wellbeing [32]. Determining if and how cities where study sites are located are different from each other can help to determine if adaptations are needed for translating interventions that aim to modify “place-dependent behaviors” (e.g., physical activity or food purchasing and consumption behaviors) across different types of cities and can also help understand in which type of city an intervention may be most effective [33,34].

The social environment of neighborhoods includes any characteristic about neighborhood-dwellers, expressed as a group-level measure (e.g., median household income, by zip code). Meanwhile, the built environment refers to any human-made neighborhood-level characteristic, such as roads, transport systems, pedestrian and cycling infrastructure, parks, buildings, and zoning and land-use.

The section below provides a general description of the geospatial area used in this study. However, a more detailed description of the operationalization and data sources used for constructing each of the neighborhood-level GIS variables to assess the social and built environments is available in Appendix A.

### 2.4. Social Environment Measures

In this study, we characterized the social environment with respect to the sociodemographic profile of neighborhoods across the study areas. Data for building social environment variables were primarily sourced from the 2018 American Community Survey [35].

The following social environment variables were calculated for each participating zip code: net population density, defined number of residents over the total area of the neighborhood with assigned residential land-use; percentage of residents by age group (number of people less than 35 years of age, between 35 and 64 years of age, and 65 years or older, residing in each zip code); predominant racial or ethnic group in the neighborhood (Asian, Black, Hispanic, White, other predominant group; mixed neighborhood (no predominant group)); percentage of the population living in poverty; predominant housing type in the zip code (single-family homes, row homes, duplex, apartments or condominiums, other types of housing, mixed neighborhood (no predominant type of housing)); percentage of vacant housing units in the neighborhood; and percentage of rented (vs. owned) housing units.

### 2.5. Built Environment Measures

In terms of the built environment, we measured physical neighborhood features within the realms of urban design, healthcare infrastructure, legal recreational drug vending outlets, recreational physical activity infrastructure, and the food environment, as all of these neighborhood environmental domains include place-based factors that can influence obesity-related health behaviors, including dietary and physical activity patterns.

#### 2.5.1. Urban Design

Urban design environment zip-code-level variables constructed in GIS for this analysis included: land-use mix (based on a measure of entropy, where a higher score on a range of 0–1 indicates higher entropy); road-network connectivity (operationalized as the density of 3- and 4-way intersections in the road network); the walkability index (as defined by Frank, et al. [36], and consisting of a combination of net residential density, road-network connectivity, and net population density); and transit stop density (number of public transit stops per kilometer squared).

#### 2.5.2. Healthcare Infrastructure

Healthcare infrastructure GIS zip-code-level variables included in this analysis were the density (i.e., number of units per 1000 inhabitants) of hospitals and other healthcare facilities, including clinics, pharmacies, and drug stores.

#### 2.5.3. Legal Drug Environment

The legal drug environment was characterized by calculating the density (units per 1000 inhabitants) of available licensed tobacco sales locations and licensed liquor stores per zip code.

#### 2.5.4. Physical Activity Environment

In terms of the physical activity environment, the following GIS zip-code-level variables were calculated to characterize the availability of public and private recreation facilities: public park density, public recreation center density, density of YMCA facilities available, and density of private fitness and recreational or sport facilities. Each of these population-density-based variables were estimated per 1000 inhabitants. Additionally, we built additional variables to characterize the walking and bicycling infrastructure in each zip code. These included the proportion of the road network with coverage by sidewalks, any bicycle lane, and protected bicycle lanes.

#### 2.5.5. Neighborhood Food Environment

Finally, we constructed the following measures to characterize the neighborhood food environment for both cities at the zip code level: density of food access points (i.e., places where one purchases food/groceries for eating at home), including counts of supermarket/grocery stores and bodegas/convenience stores per 1000 inhabitants; and food-service-point density (i.e., places where one goes to have a meal), including number of fast-food restaurants per 1000 inhabitants.

### 2.6. Individual-Level Dietary and Physical Activity Behaviors

The parent study collected baseline data on dietary and physical activity behaviors among all enrolled participants. In terms of diet, participants self-reported the number of portions of fruits and vegetables they consumed on an average day in two separate survey items derived from the Block Fruit and Vegetable Dietary Screener [37]. Based on these responses, we categorized each participant as meeting or not meeting the USDA Dietary Guidelines for Americans, which recommend consuming a minimum of five portions of fruits and vegetables per day [38]. Similarly, participants self-reported the number of portions of sugar-sweetened beverages (SSBs) they consumed daily using questions from the 2013 Centers for Disease Control and Prevention Behavioral Risk Factors Surveillance System Questionnaire (BRFSS). Because SSB intake is considered an unhealthy behavior, and the consumption of any amount of SSBs on a daily basis is considered a risk factor for obesity and diabetes, we dichotomized this variable as those who consume any SSBs on a daily basis, versus those that consume none [39].

Physical activity was measured using the short version of the International Physical Activity Questionnaire (IPAQ), which collects self-reported data on frequency (days per week) and duration (minutes per day) of participation in walking, other moderate-intensity physical activities, and vigorous-intensity physical activities during the past 7 days [40]. IPAQ data were used to derive two dichotomous variables: meeting the current US physical activity guidelines for adults of accruing 150 min per week or more of moderate- to vigorous-intensity physical activity through walking only (yes/no) [41] and meeting the US physical activity guidelines with moderate- to vigorous-intensity physical activities, excluding walking [41]. A detailed description of variable operationalization for these measures is included in the Appendix A.

### 2.7. Statistical Analysis

Descriptive statistics (counts and percentages) were calculated to describe the social and built environments of neighborhoods, defined for this study as zip codes, across New York City and Philadelphia. Fisher’s Exact tests, to account for the small sample sizes in some of the cells, were used to draw comparisons of the environmental characteristics of neighborhoods between cities and between zip codes where PGLB trial participants resided and those with no PGLB trial participants, both overall and within each city. Finally, Fisher’s Exact tests, or Chi-square tests, as appropriate depending on sample sizes per cells, were used to determine if there were statistically significant differences in dietary and physical activity behaviors among PGLB trial participants at baseline by varying levels of neighborhood environment characteristics. Analyses were performed on February 2021 and March 2023 using STATA version 16 [42].

## 3. Results

### 3.1. Sample Characteristics

The final analytic sample size for this secondary data analysis study consisted of 243 total zip codes across New York City (n = 197) and Philadelphia (n = 46). Of these, 42 corresponded to the residential location of PGLB study participants at baseline (17 in New York City, and 25 in Philadelphia). Complete physical activity and dietary data were available for 292 PGLB study participants (New York = 114; Philadelphia = 178).

As shown in Table 1, Table 2 and Table 3, our results revealed that, in both studied cities, PGLB neighborhoods were significantly more likely to have a high proportion (over 50%) of residents living in poverty than other neighborhoods of the same city. This points to an economic place-based disparity impacting people with SMI enrolled in the PGLB study. Another general commonality of the neighborhoods where PGLB participants resided at baseline was that place-based elements pertaining to the healthcare, legal drug, and food environments did not significantly vary from those across all other neighborhoods of their respective cities (i.e., disparities in access to healthcare, pharmacies, or food vending outlets impacting PGLB neighborhoods were not identified).

### 3.2. Within City Differences: PGLB Neighborhoods Versus Non-Study Neighborhoods

Significant differences between neighborhoods where PGLB study participants resided compared to all other neighborhoods of their respective cities were identified for at least one of the examined place-based indicators for the social, urban design, and physical activity environments (Table 1, Table 2 and Table 3). These within-city differences were not consistent across both examined cities. For example, while all physical activity environment variables were comparable between PGLB and non-PGLB neighborhoods in New York City (i.e., no within-city disparities in access to physical activity neighborhood resources were identified for PGLB participants), in Philadelphia, there were significantly more private fitness facilities in non-PGLB neighborhoods as compared to PGLB study neighborhoods. Likewise, PGLB participants in Philadelphia reside in neighborhoods with a significantly lower likelihood of having a high availability of hospitals than all other neighborhoods in the city of Philadelphia.

### 3.3. Between City Differences in Place-Based Factors

Table 1, Table 2 and Table 3 present findings on between- and within-city differences in place-based, neighborhood-level characteristics for the sites of the PGLB study. Several significant differences in the distribution of relevant place-based factors for lifestyle behaviors were identified between New York City and Philadelphia.

Regarding the social environment (Table 1), all place-based factors examined differed significantly across both study cities, except for the proportion of population per zip code with a college degree. Philadelphia had a significantly higher proportion of neighborhoods than New York City with low population and residential densities, younger residents, and high vacancy rates. Conversely, New York City had more mixed-race neighborhoods (i.e., neighborhoods with no predominant racial/ethnic group), and with renter-occupied units than Philadelphia.

Similar results were observed when comparing the overall distribution of place-based, neighborhood-level factors pertaining to the urban design and healthcare environments across study cities (Table 2). Neighborhood walkability, land-use mix, and public transit access differed significantly between New York City and Philadelphia, with road-network connectivity being the only comparable place-based factor of the urban design environment of both cities. Likewise, the two healthcare neighborhood environment factors examined in this study (hospital and pharmacy/drug store density) differed significantly across cities, with a significantly higher proportion of neighborhoods having high access to hospitals in Philadelphia, and lower proportion of neighborhoods having good access to pharmacies/drug stores, relative to New York City.

Moreover, the two study cities were comparable in some, but not all, of the assessed legal drug, physical activity, and food environment characteristics (Table 1 and Table 2). For instance, with regard to the legal drug environment, the distribution of tobacco retailers across neighborhoods in New York City versus Philadelphia was comparable. However, New York City had significantly more neighborhoods with medium and high access to liquor stores than Philadelphia. As for the distribution of recreational physical activity assets (e.g., parks, recreation and fitness centers, YMCAs), the two cities were comparable in all aspects expect for public park access, which was significantly higher across New York City neighborhoods. Pedestrian and bicycle infrastructure indicators were significantly higher across city neighborhoods in New York. Finally, more neighborhoods in Philadelphia had high access to bodegas and convenience stores than in New York. However, the availability of supermarkets and fast-food restaurants was comparable across both cities.

### 3.4. Baseline Differences in Dietary Behaviors and Physical Activity by Place-Based Factors, among PGLB Study Participants

#### 3.4.1. Dietary Behaviors

SSB intake did not differ significantly at baseline by neighborhood-level, place-based factors. However, we identified significant differences in the baseline levels of daily fruit and vegetable intake by place-based factors. Regarding the social environment, insufficient fruit and vegetable intake (<5 portions a day) was significantly higher among PGLB participants residing in neighborhoods where the majority of adult residents had a college degree or higher. In terms of neighborhood urban design, insufficient fruit and vegetable intake was significantly higher among those residing in neighborhoods with high connectivity and walkability. Conversely, insufficient fruit and vegetable intake was lower among those residing in neighborhoods with high access to tobacco retailers (legal drug environment). High bicycle lane coverage in the neighborhood was associated with high levels of insufficient fruit and vegetable intake (physical activity environment). No food environment characteristics were associated with differences in fruit and vegetable intake among PGLB participants at baseline.

#### 3.4.2. Physical Inactivity (Not Meeting Physical Activity Guidelines)

No significant differences by any of the studied place-based factors were observed with respect to the proportion of PGLB participants *not* meeting physical activity guidelines (>150 min per week) by walking. However, several differences by place-based factors were observed for not meeting minimum guidelines through other types of MVPA, excluding walking (i.e., exercise participation, sport, cycling). Differences in non-walking MVPA at baseline by place-based factors were identified across all types of environmental neighborhood constructs examined (social, urban design, healthcare, legal drugs, physical activity and food environments). For example, at baseline, PGLB participants residing in high-density neighborhoods were significantly more likely to not meet minimum physical activity guidelines via non-walking MVPA. Similarly, those residing in apartment homes were more likely to have insufficient levels of non-walking MVPA than PGLB participants residing in other types of housing units. PGLB participants residing in areas with high levels of housing vacancy, and in medium-to-high renter-occupied areas, were significantly more likely to not be meeting physical activity guidelines through non-walking MVPA. Likewise, urban design features such as high road-network connectivity, high walkability, and high access to transit, and higher availability of infrastructure for utilitarian (transport-based) physical activity like sidewalks and bicycle lanes were associated with higher proportions of PGLB participants not meeting physical activity guidelines via non-walking MPVA. Finally, residing in neighborhoods with high access to hospitals or areas with low access to tobacco retail facilities was associated with not meeting physical activity guidelines via MVPA. Table 4, Table 5 and Table 6 summarize the results of this analysis.

## 4. Discussion

In this study, we used an ecological approach to characterize a series of neighborhood-level placed-based factors within the context of a peer-led healthy lifestyle effectiveness trial of people with SMI living in supportive housing in New York City and Philadelphia. Our results showed significant differences in neighborhood environments between and within cities, and further identified neighborhood characteristics associated with participants’ baseline healthy eating and physical activity behaviors. Our findings underscore the importance of considering the spatial distribution of place-based factors when designing and delivering interventions that seek to modify health-related behaviors that are inherently place-dependent (e.g., physically active lifestyles require facilities to exercise, participate in sport, or engage in active travel). Our study has high relevance for lifestyle intervention work and is among the first to underscore the need to examine the influences of neighborhood environments when designing, testing, and analyzing results from healthy lifestyle interventions for people with SMI. Indeed, the effectiveness of interpersonal support for engaging in behavioral strategies (e.g., self-monitoring, goal-setting, problem-solving) in improving obesity-related behaviors among people with SMI could be modified by the underlying place-based factors of the areas where participants reside.

### 4.1. The Role of Place in the Health and Wellbeing of People with SMI: A Nascent Field

To date, most of the work demonstrating links between urban design, environmental characteristics and assets (e.g., built environment, food environment, healthcare environment) and health behaviors has been conducted among healthy adults without SMI [26,29,43,44,45,46]. Studies examining these health and place relations among specific subgroups (e.g., older adults, children, racial/ethnic minorities) have uncovered multiple differences in terms of the factors and ways (e.g., direction of the associations) in which urban environments influence health behaviors across these different groups. The ways in which the built, social, healthcare, transit, and food environment of cities influences health behaviors among people with SMI, and, in particular, among those that belong to low-income, predominantly minoritized racial and ethnic communities, remains largely understudied. The few studies that have been conducted indicate that people with SMI tend to reside in neighborhoods with unhealthy environments characterized by high crime and drug activity, a high concentration of tobacco retailers, and other indicators of social disadvantage [15,16]. The moderating role of environmental factors on health behavior interventions is gaining recognition within the general fields of physical activity and healthy eating research [17]. However, few studies of people with SMI recognize these influences through their design, statistical power, or analytic approaches.

### 4.2. The House-Poor Hypothesis and the Role of the Social Environment in Healthy Eating and Active Living among People with SMI in Cities

The study findings showed that participants with SMI residing in areas with high educational attainment among residents had lower levels of fruit and vegetable intake than those residing in neighborhoods where the community had lower levels of education. One hypothesis to explain this may be the notion of being “house poor”; that is, being of a low socioeconomic status individually but residing in an area where most people have a higher income or educational attainment may make access to certain resources too expensive [47]. For instance, the cost of shopping for healthy foods, such as fresh fruits and vegetables, in local shops in these neighborhoods, may be more expensive in areas where the majority of residents have a high level of educational attainment (which tends to be highly correlated with income), thus resulting in insufficient fruit and vegetable intake (<5 portions a day) for people living in these neighborhoods that do not have the economic means to afford these local prices. Hence, placing supportive housing units for low-income people with SMI in these high-income settings without providing financial support may lead to the unintended consequence of decreased economic access to healthy foods being available in these neighborhoods. This hypothesis warrants further exploration in future studies.

Similarly, the fact that, at baseline, participants with SMI residing in highly walkable neighborhoods had significantly lower levels of fruit and vegetable intake may be explained by other place-based factors known to be highly correlated with walkability (higher cost of living, neighborhood appreciation, etc.) [48,49]. Notably, none of the variables directly assessing the geographic food environment (presence of supermarkets, etc.) were significantly associated with fruit and vegetable intake among PGLB participants at baseline. Methodologically, this could be reflective of unaccounted confounding in our exploratory analysis (as we were not powered for this). An alternative explanation is that the observed relations may be the result of the interactions with other social determinants of health with geographic food access (e.g., the availability of healthy food options in the neighborhood may not be associated with PGLB participants’ food consumption habits as these types of stores may be unaffordable for them, or not culturally matched to their dietary preferences) [47,50]. Future research is needed to explore this hypothesis.

### 4.3. Walkability, Pedestrian and Cycling Infrastructure and Physical Activity among People with SMI: Counter-Intuitive Findings Requiring Further Study

Interestingly, meeting physical activity guidelines by walking (the most prevalent type of physical activity and the one in which most of the population can engage) did not significantly vary by any of the examined place-based factors. This is likely because both cities in this study (New York and Philadelphia) have an urban design that is highly supportive of walking, so there was insufficient environmental variability to assess possible differences in walking behaviors. Identifying the specific environmental elements and mechanisms by which place influences health in this population requires more research. Additionally, several place-based factors were associated with meeting physical activity guidelines via non-walking MVPA. However, virtually all associations can be considered counter-intuitive to what the built environment and physical activity literature for the general population has found to date: factors such as neighborhood walkability, road-network connectivity, population density, park access, or the presence of bicycle lanes are usually positively associated with physical activity (although that tends to be mainly through walking for transport behaviors). In this case, however, the observed relations with non-walking MVPA behaviors (i.e., exercising or playing sports) were inverse. One possible explanation is that this is a spurious association. However, given the consistency of results across several built environment characteristics and non-walking MVPA among our sample, this is unlikely to be the case. Another explanation might be that while these assets tend to be predictive of walking for transportation, when examined across a more variable range of geographic settings, the influence on other types of physical activity is different [34,51]. Highly dense and connected areas may offer limited opportunities for active leisure and recreation [28]. Likewise, our GIS data were not able to distinguish between high-quality and safe infrastructure (e.g., well maintained, safe parks or bicycle lanes), versus deteriorated structures. Because PGLB participants tend to reside in lower-income neighborhoods, the upkeep of resources may be sub-optimal, which may limit their use for active recreation and sport among local residents. Larger mixed-methods studies may help better elucidate the meaning of these findings. Despite this, our results highlight the need to address barriers to participation in non-walking MVPA among people with SMI residing in high-density, high-connectivity areas.

### 4.4. Considering the Broader Role of Context and Place-Based Settings When Designing and Testing Interventions to Modify Place-Based Behaviors

Another key question we sought to answer with this case study was if the spatial distribution of place-based factors was comparable across the two cities (New York and Philadelphia) where this study took place. Between-city differences are important to consider, as cities represent the macro-urban setting where the desired health behaviors take place. In the present study, we identified several between-city differences in the spatial distribution of place-based factors that, per the ecological model, may influence physical activity and/or healthy eating behaviors. As reported elsewhere, PGLB outcomes differed by study site [19]. The two supportive housing agency sites in Philadelphia reported null findings between the PGLB and usual care conditions in the study’s main health outcomes, while in the New York City site, PGLB significantly outperformed usual care on weight loss and CVD risk-reduction outcomes [19]. These site differences in study outcomes and place-based factors suggest that when conducting multisite studies, such as PGLB, a pooled analysis which combines data from across cities would not be ideal, as intervention effects may vary across cities. Our findings suggest that stratified analysis by city should be considered or if a pooled analysis is deemed necessary (to estimate “main effects” for the intervention), it is recommended to account for the clustering effect of the macro-environment (city) in the analysis.

We also found within-city differences, meaning that the neighborhoods from which PGLB study participants were drawn were found to be significantly different in terms of their physical activity, food, transit, and/or healthcare environments when compared to the other neighborhoods within the same city. This, in combination with the fact that significant relationships between neighborhood-level factors and some key behavioral outcomes of the intervention (e.g., fruit and vegetable intake, non-walking-based MVPA) were observed, suggests the limited generalizability of intervention findings to people with SMI residing in very different neighborhood environments to those of the PGLB study participants. However, this should not minimize the value of the PGLB study (and of similar interventions that recruit participants through pre-identified urban housing sites for low-income people with SMI). A high number of low-income, racial and ethnic minoritized people with SMI reside in supportive housing agencies such as the ones studied in PGLB. Carefully considering the geographic placement of these types of housing sites for racial and ethnically diverse and low-income people with SMI, especially as this relates to the availability of certain types of neighborhood assets and social factors, should be given much stronger consideration by the decision-makers and funding organizations supporting these types of facilities. This is especially relevant given that obesity, and its co-morbidities, are a highly prevalent health issue among supportive housing residents [52,53,54].

### 4.5. Limitations and Strengths

Our study has some limitations that warrant consideration. The parent study (PGLB trial) did not use a random sampling approach, but rather, selected supportive housing sites based on convenience (using pre-existing relationships to facilitate the research). This is common in behavioral research, and particularly for work with people with SMI. However, this restricted the sample size of residential zip codes (neighborhoods) among participants, precluding us from having sufficient power to conduct more sophisticated statistical analyses examining the associations of place-based factors with obesity-related behaviors at baseline among the sample (e.g., multivariable regression modeling). As such, the findings from the Chi-Square and Fisher’s Exact tests reported should be considered preliminary. All participants enrolled in the study had overweight or obesity at baseline. Hence, the relationships between place-based factors and behavioral outcomes observed at baseline may not be reflective of people with SMI who have healthy weight status. Further, the parent study considered a BMI value of 25 or above as indicative of having overweight or obesity, a criterion used for eligibility purposes, even though new studies have shown that the categorization of overweight based on a cut-point of BMI ≥ 25 is not accurate for all population groups. This study was restricted to two cities, limiting the geospatial variability of the various environmental features examined. Different relationships may be uncovered in larger studies inclusive of a broader range of cities. Finally, neighborhoods were operationalized by examining place-based factors within zip codes. A preferable approach would have been to develop individual-level, participant-centric indicators of place-based factors (with buffers of a given radius around each participant’s address). However, this study did not collect participants’ exact addresses, and a sampling strategy that included supportive housing agencies with congregate housing options (i.e., clustering participants to reside within a subset of buildings) would not have yielded added variability.

Despite these limitations, our study also has many strengths. We are among the first to use a socio-ecological lens to examine the methodological implications of sampling strategies for behavioral interventions for people with SMI by characterizing the spatial distribution of multi-level place-based factors for the PGLB study. The analysis, with respect to the location of housing sites for the PGLB study, will help to inform further analyses and follow-up studies for PGLB. Additionally, our analyses unveiled several important lessons with broader implications for behavioral intervention research at large and for people with SMI.

### 4.6. Future Research Recommendations

When designing behavioral intervention trials to improve healthy eating and physical activity (among any type of population group), investigators should seek to balance their samples across arms (experimental vs. control) by place-based factors, in the same way they would expect randomization to distribute confounders evenly. When random spatial sampling is not feasible, other strategies (i.e., spatially stratified sampling; meeting spatial quotas) should be implemented at the design phase, or during data analysis (e.g., propensity score-matching, adjusting for place-based confounders). Ideally, behavioral interventions targeting place-dependent outcomes (e.g., physical activity or healthy eating) should be powered and designed to examine potential moderation by neighborhood-level place-based factors. Understanding the role that neighborhood placed-based factors play in the health of people with SMI is critical for eliminating health inequities in these historically marginalized populations.

## 5. Conclusions

This study sought to demonstrate the importance of considering and comprehending the role of neighborhood-level place-based factors when designing, testing and analyzing behavioral interventions to improve the health of people with SMI residing in supportive housing. This study demonstrates that the physical and social characteristics of the neighborhoods (e.g., the built, social, healthcare, transit, and food environments) where people with SMI reside and participate in behavioral interventions affect the efficacy of the interventions. Statistically significant differences exist at the neighborhood level within and between cities, highlighting the need for interventions to consider the geographic location of housing and service facilities for the population being served. This is particularly significant concerning the availability of certain neighborhood assets and social factors that are statistically significant in relation to the assessed dietary and physical inactivity behaviors. As some counterintuitive results were found regarding certain healthy behaviors promoted as part of the peer-led intervention, future research must account for potential barriers to adopting a healthy lifestyle. These considerations are of the utmost importance for individuals with SMI who reside in low-income, high-density, and high-connectivity areas, where access to resources that promote healthy living may be limited and inadequate, exacerbating health disparities. The findings of this study have important implications for policy and practice, and are relevant to mental health practitioners and decision-makers working in the space of supportive housing agencies that serve low-income, racial, and ethnic minoritized individuals with SMI and, importantly, to urban planners. The consideration of health as a key component of optimal urban design and planning is not a new concept. For example, the concept of “health in all policies” approach [55], and the “8 best investments that work for promoting physical activity” [56], include the design of cities that favor active travel and that have ample built-in opportunities for active recreation for all. Some urban design initiatives for health argue that by designing cities that are inclusive, safe, and conducive to the “healthy choice” for the most vulnerable members of society (e.g., children, older adults, people with physical disabilities), we will be designing healthy, equitable cities for all [57,58,59]. Our study underscores that an often overlooked group to consider when making policy and planning urban decisions are people with SMI. Our study results support that ultimately, incorporating a place-based perspective into lifestyle interventions for individuals with SMI could increase the effectiveness of these interventions and improve health outcomes for this vulnerable population.

## Figures and Tables

**Table 1 ijerph-20-05679-t001:** Social environment characteristics of the Peer-led Group Lifestyle Balance Program (PGLB) study and city-wide neighborhoods (zip codes) of New York City, NY and Philadelphia, PA (N = 243; 2015).

	New York City			Philadelphia			Between-City Differences		
	PGLB Neighborhoods (n = 17)	Rest of the City (n = 180)	*p*-Value	PGLB Neighborhoods (n = 25)	Rest of the City (n = 21)	*p*-Value	New York City (n = 197)	Philadelphia (n = 46)	*p*-Value
	% (n)	% (n)		% (n)	% (n)		% (n)	% (n)	
Net Population Density ^1^									
Low (<20,891.76)	17.65 (3)	28.89 (52)	**0.000**	48 (12)	66.67 (14)	0.456	27.92 (55)	56.52 (26)	**0.000**
Medium (20,891.76–<58,532.75	0	35.56 (64)		44 (11)	28.57 (6)		32.49 (64)	36.96 (17)	
High (≥58,532.75)	82.35 (14)	35.56 (64)		8 (2)	4.76 (1)		39.59 (78)	6.52 (3)	
Age group									
Young (<35 years)	29.41 (5)	27.78 (50)	0.999	64 (16)	47.62 (10)	0.372	27.92 (55)	56.52 (26)	**0.001**
Middle age (35–64 years)	70.59 (12)	71.67 (129)		36 (9)	52.38 (11)		71.57 (141)	43.48 (20)	
Old (65+ years)	0	0.56 (1)		0	0		0.51 (1)	0	
Predominant race ^2^									
Asian	0	1.11 (2)	0.265	0	0	0.309	1.02 (2)	0	**0.001**
Black	5.88 (1)	8.89 (16)		28 (7)	28.57 (6)		8.63 (17)	28.26 (13)	
Hispanic/Latino	25.53 (4)	8.89 (16)		0	0		10.15 (20)	0	
White	11.76 (2)	27.78 (50)		24 (6)	42.86 (9)		26.40 (52)	39.13 (18)	
Other race ^2a^	0	0		0	0		0	0	
Mixed	58.82 (10)	53.33 (96)		28.57 (6)	48 (12)		53.81 (106)	39.13 (18)	
Population living poverty ^3^									
<50% living in poverty	64.71 (11)	91.11 (164)	**0.001**	90.48 (19)	56 (14)	**0.019**	88.89 (175)	71.74 (33)	**0.001**
≥50% living in poverty	35.29 (6)	8.89 (16)		9.52 (2)	44 (11)		11.17 (22)	28.26 (13)	
Predominant housing type ^4^									
Single Family homes	0	3.33 (6)	**0.019**	0	0	0.296	3.05 (6)	0	**0.000**
Row or single family attached	0	0		56 (14)	33.33 (7)		0	45.65 (21)	
Duplex	0	0		0	0		0	0	
Apartment (3+ units)	94.12 (16)	59.44 (107)		8 (2)	9.52 (2)		62.44 (123)	8.70 (4)	
Other ^4a^	0	0		0	0		0		
Mixed	5.88 (1)	37.22 (67)		36 (9)	57.14 (12)		34.52 (68)	45.65 (21)	
Population with a college degree									
<60% with college degree	35.29 (6)	36.1 (65)	0.999	56 (14)	38.10 (8)	0.253	36.04 (71)	47.83 (22)	0.177
≥60% with college degree	64.71 (11)	36.89 (115)		44 (11)	61.90 (13)		63.96 (126)	52.17 (24)	
Housing vacancy in the neighborhood									
Low vacancy (<7%)	47.06 (8)	37.22 (67)	0.625	12 (3)	19.05 (4)	0.782	38.07 (75)	15.22 (7)	**0.000**
Medium vacancy (7–<11.15%)	35.29 (6)	34.44 (62)		28 (7)	28.57 (6)		34.52 (68)	28.26 (13)	
High vacancy (≥11.15%)	17.65 (3)	28.33 (51)		60 (15)	52.38 (11)		27.41 (54)	56.52 (26)	
Percentage of rented (vs. owned) housing units in the neighborhood									
Low (<43%)	5.88 (1)	28.33 (51)	**0.024**	60 (15)	66.67 (14)	0.666	26.40 (52)	63.04 (29)	**0.000**
Medium (43–<63.3%)	23.53 (4)	34.44 (62)		32 (8)	33.33 (7)		33.50 (66)	32.61 (15)	
High (≥63.3%)	70.59 (12)	37.22 (67)		8 (2)	0		40.10 (79)	4.35 (2)	

^1^ Measured as inhabitants per Km^2^ of residential land-use area. ^2^ If 60% or more of residents are of a given race, neighborhood is said to be predominantly of that race/ethnicity; otherwise, it is said to be mixed. ^2a^ Other includes: Native Hawaiian, American Indian/Alaskan Native, and two or more races; the other category is as defined by the ACS. ^3^ Those whose income is ≤125% of the poverty level qualify for all Federal Assistantship Programs; those whose income is 126–185% of the poverty level qualify for some Federal Assistantship programs, and those whose income > 185% of the poverty level do not qualify for any Federal Assistantship Program. ^4^ If 60% or more of housing units are of a given type, it is said to be predominantly of that housing type; otherwise, it is said to be mixed. ^4a^ Other includes trailer or mobile homes and other types of private home (boat, RV, van, etc.). Bolded numbers indicate statistical significance at the *p* < 0.05 level.

**Table 2 ijerph-20-05679-t002:** Characteristics of the Urban Design, Health Care and Legal Drugs Environments of the Peer-led Group Lifestyle Balance Program (PGLB) study and city-wide neighborhoods (zip codes) of New York City, NY and Philadelphia, PA (N = 243; 2015).

	New York City			Philadelphia			Between-City Differences		
	PGLB Neighborhoods (n = 17)	Rest of the City (n = 180)	*p*-Value	PGLB Neighborhoods (n = 25)	Rest of the City (n = 21)	*p*-Value	New York City (n = 197)	Philadelphia (n = 46)	*p*-Value
	% (n)	% (n)		% (n)	% (n)		% (n)	% (n)	
**Urban Design Environment**									
Land Use Mix ^1^									
Low (<0.67)	11.76 (2)	41.11 (74)	**0.035**	12 (3)	9.52 (2)	0.848	38.58 (76)	10.87 (5)	**0.000**
Medium (0.67–<0.82)	41.18 (7)	32.22 (58)		32 (8)	38.10 (8)		32.99 (65)	34.78 (16)	
High (≥0.82)	47.06 (8)	26.67 (48)		56 (14)	52.38 (11)		28.43 (56)	54.35 (25)	
Connectivity ^2^									
Low (< 269)	29.41 (5)	31.11 (56)	0.412	36 (9)	52.38 (11)	0.196	30.96 (61)	43.48 (20)	0.120
Medium (269–< 363.7)	23.53 (4)	37.22 (67)		32 (8)	9.52 (2)		36.04 (71)	21.74 (10)	
High (≥ 363.7)	47.06 (8)	31.67 (57)		32 (8)	38.10 (8)		32.99 (65)	34.78 (16)	
Walkability Index ^3^									
Low (<0.37)	11.76 (2)	28.33 (51)	0.311	56 (14)	66.67 (14)	0.771	26.90 (53)	60.87 (28)	**0.000**
Medium (0.37–< 1.41)	47.06 (8)	34.44 (62)		28 (7)	19.05 (4)		35.53 (70)	23.91 (11)	
High (≥ 1.41)	41.18 (7)	37.22 (67)		16 (4)	14.29 (3)		37.56 (74)	15.22 (7)	
Transit Stops Density ^a^									
Low (<22.08)	29.41 (5)	38.89 (70)	0.559	8 (2)	19.05 (4)	0.075	38.07 (75)	33.04 (6)	**0.000**
Medium (22.08–<36.45)	35.29 (6)	63.11 (65)		12 (3)	33.33 (7)		36.04 (71)	21.74 (10)	
High (≥ 36.45)	35.29 (6)	25 (45)		80 (20)	47.62 (10)		28.89 (51)	35.22 (30)	
**Healthcare Environment**									
Hospital density ^b^									
Low (0)	29.41 (5)	42.78 (77)	0.104	0	0	**0.024**	41.62 (82)	0	**0.000**
Medium (>0–<0.000055)	52.94 (9)	26.67 (48)		52 (1)	47.62 (10)		28.93 (57)	50 (23)	
High (≥0.000055)	17.65 (3)	30.56 (55)		48 (12)	52.38 (11)		29.44 (58)	50 (23)	
Pharmacy & drug store density ^b^									
Low (<0.00014)	17.65 (3)	19.44 (35)	0.573	92 (23)	95.24 (20)	0.999	19.29 (38)	93.48 (43)	**0.000**
Medium (0.00014–<0.0003)	29.41 (5)	41.11 (74)		4 (1)	4.76 (1)		40.10 (79)	4.35 (2)	
High (≥0.0003)	52.94 (9)	39.44 (71)		4 (1)	0		40.61 (80)	2.17 (1)	
**Legal Drug Environment**									
Licensed tobacco sale place density ^b^									
Low (0)	52.94 (9)	54.44 (98)	0.298	60 (15)	66.67 (14)	0.666	54.31 (107)	63.04 (29)	0.293
Medium (>0–<0.00002)	23.53 (4)	11.11 (20)		8 (2)	0		12.18 (24)	4.35 (2)	
High (≥0.00002)	23.53 (4)	34.44 (62)		32 (8)	33.33 (7)		33.50 (66)	32.61 (15)	
Liquor store density ^b^									
Low (<0.0001)	23.53 (4)	26.67 (48)	0.278	64 (16)	61.90 (13)	0.999	26.40 (52)	63.04 (29)	**0.000**
Medium (0.0001–<0.0002)	52.94 (9)	33.89 (61)		24 (6)	23.81 (5)		35.53 (70)	23.91 (11)	
High (≥0.0002)	23.53 (4)	39.44 (71)		12 (3)	14.29 (3)		38.07 (75)	13.04 (6)	

^1^ Defined as: −[Σi (pi)∗(ln pi)]/(ln k), where p = proportion of total land uses, i = land use category, ln = natural logarithm, k = number of land uses. Range is 0–1; ^2^ Defined as: Intersection density defined as the number of 3- and 4-way intersections in a buffer area/total buffer area in squared kilometers; ^3^ Defined as: z-scored net population density + 2(z-scored connectivity ^2^) + z-scored land-use mix ^1^; ^a^ Density is area-based; ^b^ density is population-based. Bolded numbers indicate statistical significance at the *p* < 0.05 level.

**Table 3 ijerph-20-05679-t003:** Characteristics of the Physical Activity and Food Environments of the Peer-led Group Lifestyle Balance Program (PGLB) study and city-wide neighborhoods (zip codes) of New York City, NY and Philadelphia, PA (N = 243; 2015).

	New York City			Philadelphia			Between-City Differences		
	PGLB Neighborhoods (n = 17)	Rest of the City (n = 180)	*p*-Value	PGLB Neighborhoods (n = 25)	Rest of the City (n = 21)	*p*-Value	New York City (n = 197)	Philadelphia (n = 46)	*p*-Value
	% (n)	% (n)		% (n)	% (n)		% (n)	% (n)	
**Physical Activity Environment**									
*Public and private recreation facilities*									
Public park density ^b^									
Low (<0.000211)	5.88 (1)	21.67 (39)	0.191	96 (24)	80.95 (17)	0.163	20.30 (40)	89.13 (41)	**0.000**
Medium (0.000211–<0.00632)	35.29 (6)	38.89 (70)		4 (1)	19.05 (4)		38.58 (76)	10.87 (5)	
High (≥0.00632)	58.82 (10)	39.44 (71)		0	0		41.12 (81)	0	
Public Recreation centre density ^b^									
Low (<0.00002)	29.41 (5)	30.56 (55)	0.069	52 (13)	38.10 (8)	0.596	30.46 (60)	45.65 (21)	0.103
Medium (0.00002–<0.00005)	58.82 (10)	33.89 (61)		16 (4)	28.57 (6)		36.04 (71)	21.74 (10)	
High (≥0.00005)	11.76 (2)	35.56 (64)		32 (8)	33.33 (7)		33.50 (66)	32.61 (15)	
YMCA centre density ^b^									
Low (0)	100 (17)	88.89 (160)	0.226	92 (23)	85.71 (18)	0.648	89.85 (177)	89.13 (41)	0.794
Medium (NA)	0	0		0	0		0	0	
High (>0)	0	11.11 (20)		8 (2)	14.29 (3)		10.15 (20)	10.87 (5)	
Private fitness and recreation sport centre density ^b^									
Low (<0.17)	23.53 (4)	41.67 (75)	0.235	36 (9)	52.38 (11)	**0.035**	40.10 (79)	43.48 (20)	0.799
Medium (0.17–<1.56)	41.18 (7)	25.56 (46)		36 (9)	4.76 (1)		26.90 (53)	21.74 (10)	
High (≥1.56)	35.29 (6)	32.78 (59)		28 (7)	42.86 (9)		32.99 (65)	34.78 (16)	
*Walking and cycling infrastructure*									
Sidewalk coverage ^1^									
Low (<1.11)	52.94 (9)	34.44 (62)	0.346	4 (1)	42.86 (9)	**0.027**	36.04 (71)	21.74 (10)	**0.027**
Medium (1.11–<1.334)	29.41 (5)	35 (63)		32 (8)	23.81 (5)		34.52 (68)	28.26 (13)	
High (≥1.334)	17.65 (3)	30.56 (55)		64 (16)	33.33 (16)		29.44 (58)	50 (23)	
Bicycle lanes coverage ^2^									
Low (<0.05)	11.76 (2)	40.56 (73)	**0.013**	8 (2)	19.05 (4)	0.417	38.07 (75)	13.04 (6)	**0.001**
Medium (0.05–0.12)	23.53 (4)	28.89 (52)		52 (13)	57.14 (12)		28.43 (56)	54.35 (25)	
High (≥0.12)	64.71 (11)	30.56 (55)		40 (10)	23.81 (5)		33.50 (66)	32.61 (15)	
Protected bicycle lane coverage ^3^									
Low (<0.007)	0	28.89 (52)	**0.001**	88 (22)	90.48 (19)	0.999	26.40 (52)	89.13 (41)	**0.000**
Medium (0.007–<0.27)	17.65 (3)	33.89 (61)		12 (3)	9.52 (2)		32.49 (64)	10.87 (5)	
High (≥0.27)	82.35 (14)	37.22 (67)		0	0		41.12 (81)	0	
**Food Environment**									
*Food Access points*									
Supermarket and grocery stores density ^b^									
Low (<0.67)	35.29 (6)	33.33 (60)	0.903	24 (6)	42.86 (9)	0.365	33.50 (66)	32.61 (15)	0.371
Medium (0.67–<1.11)	26.41 (5)	35.56 (64)		32 (8)	19.05 (4)		35.03 (69)	26.09 (12)	
High (≥1.11)	35.29 (6)	31.11 (56)		44 (11)	38.10 (8)		31.47 (62)	41.30 (19)	
Bodegas, convenience and general stores density ^b^									
Low (<0.08)	52.94 (9)	37.22 (67)	0.132	12 (3)	14.29 (3)	0.999	38.58 (76)	13.04 (6)	**0.000**
Medium (0.08–<0.15)	41.18 (7)	36.67 (66)		16 (4)	14.29 (3)		37.06 (73)	15.22 (7)	
High (≥0.15)	5.88 (1)	26.11 (47)		72 (18)	71.43 (15)		24.37 (48)	71.74 (33)	
*Food service points*									
Fast Food restaurant density ^b^									
Low (<0.40)	25.53 (4)	30.56 (55)	0.597	56 (14)	38.10 (8)	0.151	29.95 (59)	47.83 (22)	0.075
Medium (0.40–<0.61)	47.06 (8)	33.33 (60)		24 (6)	33.33 (7)		34.52 (68)	28.26 (13)	
High (≥0.61)	29.41 (5)	36.11 (65)		5 (20)	28.57 (6)		35.53 (70)	23.91 (11)	

^1^ Defined as: total length in linear kms of sidewalks/total linear kms of roads within the zip code. ^2^ Defined as: total length in linear kms of bicycle lenes/total linear kms of roads within the zip code. ^3^ Defined as: total length in linear kms of protected bicycle lanes/total linear kms of roads within the zip code. ^b^ Density is population-based and estimated per 1000 inhabitants. Bolded numbers indicate statistical significance at the *p* < 0.05 level.

**Table 4 ijerph-20-05679-t004:** Baseline frequency of intake of sugar-sweetened beverages, fruits and vegetables, and physical activity levels among participants of the Peer-led Group Lifestyle Balance Program (PGLB), by neighbourhood social environment characteristics (n = 292).

	Dietary Behaviors				Physical Activity			
	<5 Portions of F&V per Day (n = 210)	*p*-Value	≥1 Portions of SSB per Day(n = 189)	*p*-Value	<150 min of Walking per Week (n = 116)	*p*-Value	<150 min of MVPA per Week ^0^ (n = 181)	*p*-Value
	% (n)		% (n)		% (n)		% (n)	
Net Population Density ^1^								
Low (<20,891.76)	26.19 (55)	0.434	30.16 (57)	0.184	27.59 (32)	0.876	22.65 (41)	**0.022**
Medium (20,891.76–<58,532.75	23.33 (49)		23.81 (45)		20.69 (24)		22.10 (40)	
High (≥58,532.75)	50.48 (106)		46.03 (87)		51.72 (60)		55.25 (100)	
Age group								
Young (<35 years)	57.14 (120)	0.978	57.67 (109)	0.822	56.90 (66)	0.934	54.70 (99)	0.271
Middle age (35–64 years)	42.86 (90)		42.33 (80)		43.10 (50)		45.30 (82)	
Old (65+ years)	0.00 (0)		0.00 (0)		0.00 (0)		0.00 (0)	
Predominant race ^2^								
Asian	0.00 (0)	0.180	0.00 (0)	0.771	0.00 (0)	0.939	0.00 (0)	0.402
Black	19.52 (41)		22.22 (42)		24.14 (28)		19.34 (35)	
Hispanic/Latino	2.38 (5)		2.12 (4)		2.59 (3)		3.31 (6)	
White	22.86 (48)		21.69 (41)		19.83 (23)		20.44 (37)	
Other race ^2a^	55.24 (116)		0.00 (0)		0.00 (0)		0.00 (0)	
Mixed			53.97 (102)		55.45 (62)		56.91 (103)	
Population living poverty ^3^								
<50% living in poverty	61.43 (129)	0.649	58.73 (111)	0.372	58.62 (68)	0.571	64.09 (116)	0.121
≥50% living in poverty	38.57 (81)		41.27 (78)		41.38 (48)		35.91 (65)	
Predominant housing type ^4^								
Single Family homes	0.00 (0)	0.462	0.00 (0)	0.160	0.00 (0)	0.679	0.00 (0)	**0.000**
Row or single family attached	14.76 (31)		18.52 (35)		14.66 (17)		9.39 (17)	
Duplex	0.00 (0)		0.00 (0)		0.00 (0)		0.00 (0)	
Apartment (3+ units)	55.71 (117)		50.79 (96)		57.76 (67)		62.43 (113)	
Other ^4a^	0.00 (0)		0.00 (0)		0.00 (0)		0.00 (0)	
Mixed	29.52 (62)		30.69 (58)		27.59 (32)		28.18 (51)	
Population with a college degree								
<60% with college degree	25.71 (54)	**0.005**	29.63 (56)	0.669	32.75 (38)	0.492	27.07 (49)	0.106
≥60% with college degree	74.29 (156)		70.37 (133)		67.24 (78)		72.93 (132)	
Housing vacancy in the neighborhood								
Low vacancy (<7%)	10 (21)	0.861	7.94 (15)	0.054	8.62 (10)	0.560	13.26 (24)	**0.021**
Medium vacancy (7–<11.15%)	26.19 (55)		24.34 (46)		28.45 (33)		29.28 (53)	
High vacancy (≥11.15%)	63.81 (134)		67.72 (128)		62.93 (73)		57.46 (104)	
Percentage of rented (vs. owned) housing units in the neighborhood								
Low (<43%)	14.76 (31)	0.884	17.46 (33)	0.068	12.07 (14)	0.390	7.73 (14)	**0.000**
Medium (43–<63.3%)	44.29 (93)		46.56 (88)		43.97 (51)		43.65 (79)	
High (≥63.3%)	40.95 (86)		35.98 (68)		43.97 (51)		48.62 (88)	

^0^ Excludes walking.^1^ Measured as inhabitants per Km^2^ of residential land-use area. ^2^ If 60% or more of residents are of a given race, neighborhood is said to be predominantly of that race/ethnicity; otherwise, it is said to be mixed. ^2a^ Other includes: Native Hawaiian, American Indian/Alaskan Native, and two or more races in the other category as defined by the ACS. ^3^ Those whose income is ≤125% of the poverty level qualify for all Federal Assistantship Programs; those whose income is 126–185% of the poverty level qualify to some Federal Assistantship programs; and those whose income >185% of the poverty level do not qualify for any Federal Assistantship Program. ^4^ If 60% or more of housing units are of a given type it is said to be predominantly of that housing type; otherwise, it is said to be mixed. ^4a^ Other includes: trailer or mobile homes and other types of private home (boat, RV, van, etc.). Bolded numbers indicate statistical significance at the *p* < 0.05 level.

**Table 5 ijerph-20-05679-t005:** Baseline frequency of intake of sugar-sweetened beverages, fruits and vegetables, and physical activity levels among participants of the Peer-led Group Lifestyle Balance Program (PGLB), by neighbourhood urban design, healthcare, and legal drug environment characteristics (n = 292).

	Dietary Behaviors				Physical Activity			
	<5 Portions of F&V per Day (n = 210)	*p*-Value	≥1 Portions of SSB Per Day (n = 189)	*p*-Value	<150 min of Walking per Week (n = 116)	*p*-Value	<150 min of MVPA Per Week ^0^ (n = 181)	*p*-Value
	% (n)		% (n)		% (n)		% (n)	
**Urban Design Environment**								
Land Use Mix ^1^								
Low (<0.67)	16.67 (35)	0.125	18.52 (35)	0.542	22.41 (26)	0.597	20.99 (38)	0.555
Medium (0.67–<0.82)	39.52 (83)		35.98 (68)		36.21 (42)		38.12 (69)	
High (≥0.82)	43.81 (92)		45.50 (86)		41.38 (48)		40.88 (74)	
Connectivity ^2^								
Low (<269)	21.43 (45)	**0.047**	24.34 (46)	0.167	26.72 (31)	0.325	19.89 (36)	**0.017**
Medium (269–<363.7)	16.19 (34)		17.99 (34)		11.21 (13)		14.92 (27)	
High (≥363.7)	62.38 (131)		57.67 (109)		62.07 (62.07)		65.19 (118)	
Walkability Index ^3^								
Low (<0.37)	26.67 (56)	**0.038**	31.22 (59)	0.860	29.31 (34)	0.873	23.20 (42)	**0.003**
Medium (0.37–<1.41)	25.71 (54)		26.46 (50)		27.59 (32)		29.28 (53)	
High (≥1.41)	47.62 (100)		42.33 (80)		43.10 (50)		47.51 (86)	
Transit Stops Density ^a^								
Low (<22.08)	8.10 (17)	0.840	6.35 (12)	0.472	9.48 (11)	0.132	10.50 (199	**0.015**
Medium (22.08–<36.45)	18.57 (39)		17.99 (34)		23.28 (27)		20.99 (38)	
High (≥36.45)	73.33 (154)		75.66 (143)		67.24 (78)		68.51 (124)	
**Healthcare Environment**								
Hospital density ^b^								
Low (0)	6.67 (14)	0.059	5.29 (10)	0.339	6.90 (8)	0.886	9.39 (17)	**0.008**
Medium (>0–<0.000055)	44.29 (93)		46.56 (88)		49.14 (57)		49.17 (89)	
High (≥0.000055)	49.05 (103)		48.15 (91)		43.97 (51)		41.44 (75)	
Pharmacy and drug stores density ^b^								
Low (<0.00014)	57.14 (120)	0.609	59.26 (112)	0.553	57.76 (67)	0.978	53.04 (96)	0.114
Medium (0.00014–<0.0003)	14.29 (30)		15.34 (29)		15.52 (18)		15.47 (28)	
High (≥0.0003)	28.57 (60)		25.40 (48)		26.72 (31)		31.49 (57)	
**Legal Drug Environment**								
Licensed tobacco sale places density ^b^								
Low (0)	65.24 (137)	**0.015**	62.43 (118)	0.215	60.34 (70)	0.911	69.61 (126)	**0.000**
Medium (>0–<0.00002)	6.67 (14)		7.41 (14)		10.34 (12)		10.50 (19)	
High (≥0.00002)	28.10 (59)		30.16 (57)		29.31 (34)		19.89 (36)	
Liquor stores density ^b^								
Low (<0.0001)	27.62 (58)	0.148	32.80 (62)	0.310	29.45 (33)	0.771	23.20 (42)	**0.002**
Medium (0.0001–<0.0002)	49.52 (104)		47.62 (90)		52.59 (61)		51.93 (94)	
High (≥0.0002)	22.86 (48)		19.58 (37)		18.97 (22)		24.86 (45)	

^0^ Excludes walking. ^1^ Defined as:−[Σi (pi)∗(ln pi)]/(ln k), where p = proportion of total land uses, i = land-use category, ln = natural logarithm, k = number of land uses. Range is 0–1. ^2^ Defined as: intersection density defined as the number of 3- and 4-way intersections in a buffer area/total buffer area in squared kilometers. ^3^ Defined as: z-scored net population density + 2(z-scored connectivity ^2^) + z-scored land-use mix ^1^. ^a^ Density is area-based. ^b^ Density is population-based and estimated per 1000 inhabitants. Bolded numbers indicate statistical significance at the *p* < 0.05 level.

**Table 6 ijerph-20-05679-t006:** Baseline frequency of intake of sugar-sweetened beverages, fruits and vegetables, and physical activity levels among participants of the Peer-led Group Lifestyle Balance Program (PGLB), by neighbourhood food and physical activity environment characteristics (n = 292).

	Dietary Behaviors				Physical Activity			
	<5 Portions of F&V per Day (n = 210)	*p*-Value	≥1 Portions of SSB per Day (n = 189)	*p*-Value	<150 min of Walking per Week (n = 116)	*p*-Value	<150 min of MVPA per Week ^0^ (n = 181)	*p*-Value
	% (n)		% (n)		% (n)		% (n)	
**Physical Activity Environment**								
*Public and private recreation facilities*								
Public parks density ^b^								
Low (<0.000211)	65.71 (138)	0.432	70.37 (133)	0.226	66.38 (77)	0.909	58.01 (105)	**0.000**
Medium (0.000211–<0.00632)	14.29 (30)		11.64 (22)		13.79 (16)		16.57 (30)	
High (≥0.00632)	20 (42)		17.99 (34)		19.83 (23)		25.41 (46)	
Public Recreation centres density ^b^								
Low (<0.00002)	29.05 (61)	0.226	32.28 (61)	0.845	27.59 (32)	0.173	26.52 (48)	**0.029**
Medium (0.00002–<0.00005)	38.57 (81)		37.04 (70)		34.48 (40)		40.88 (74)	
High (≥0.00005)	32.38 (68)		30.69 (58)		37.93 (44)		32.60 (59)	
YMCA centres density ^b^								
Low (0)	95.71 (201)	0.515	95.77 (181)	0.543	93.10 (108)	0.172	94.48 (171)	0.456
Medium (NA)	0		0		0		0	
High(>0)	4.29 (9)		4.23 (8)		6.90 (8)		5.52 (10)	
Private fitness and recreation sport centres density ^b^								
Low (<0.17)	27.14 (57)	0.298	23.34 (46)	0.729	31.03 (36)	0.141	24.86 (45)	**0.045**
Medium (0.17–<1.56)	35.71 (75)		39.68 (75)		33.62 (39)		33.15 (60)	
High (≥1.56)	37.14 (78)		35.98 (68)		35.32 (41)		41.99 (76)	
*Walking and cycling infrastructure*								
Sidewalk coverage ^1^								
Low (<1.11)	11.43 (24)	0.051	9.52 (18)	0.433	12.93 (15)	0.608	13.81 (25)	**0.004**
Medium (1.11–<1.334)	48.10 (101)		43.39 (82)		44.83 (52)		48.62 (88)	
High (≥1.334)	40.48 (85)		47.09 (89)		42.24 (49)		37.57 (68)	
Bicycle lanes coverage ^2^								
Low (<0.05)	5.24 (11)	**0.002**	5.29 (10)	0.823	7.76 (9)	0.324	6.63 (12)	**0.000**
Medium (0.05–.12)	31.43 (66)		38.62 (73)		34.48 (40)		28.73 (52)	
High (≥0.12)	63.33 (133)		56.08 (106)		57.76 (67)		64.64 (117)	
Protected bicycle lane coverage ^3^								
Low (<0.007)	59.05 (124)	0.157	62.43 (118)	0.290	56.03 (65)	0.663	48.07 (87)	**0.000**
Medium (0.007–<0.27)	10 (21)		11.64 (22)		12.93 (15)		13.81 (25)	
High (≥0.27)	30.95 (65)		25.93 (49)		31.03 (36)		39.12 (69)	
**Food Environment**								
*Food Access points*								
Supermarket and grocery stores density ^b^								
Low (<0.67)	18.57 (39)	0.750	18.99 (34)	0.353	18.97 (22)	0.273	23.76 (43)	0.061
Medium (0.67–<1.11)	40.48 (85)		42.33 (80)		34.48 (40)		38.12 (69)	
High (≥1.11)	40.95 (86)		39.68 (75)		46.55 (54)		38.12 (69)	
Bodegas, convenience and general stores density ^b^								
Low (<0.08)	37.62 (79)	0.435	34.92 (66)	0.233	38.79 (45)	0.634	44.20 (80)	**0.000**
Medium (0.08–<0.15)	19.05 (40)		16.40 (31)		17.24 (20)		22.65 (41)	
High (≥0.15)	43.33 (91)		48.68 (92)		43.97 (51)		33.15 (60)	
*Food service points*								
Fast Food restaurant density ^b^								
Low (<0.40)	29.52 (62)	0.987	33.33 (63)	0.197	26.72 (31)	0.300	21.55 (39)	**0.000**
Medium (0.40–<0.61)	39.52 (83)		37.04 (70)		44.83 (52)		45.30 (82)	
High (≥0.61)	30.95 (65)		29.63 (56)		28.45 (33)		33.15 (60)	

^0^ Excludes walking. ^1^ Defined as: total length in linear kms of sidewalks/total linear kms of roads within the zip code. ^2^ Defined as: total length in linear kms of bicycle lenes/total linear kms of roads within the zip code. ^3^ Defined as: total length in linear kms of protected bicycle lanes/total linear kms of roads within the zip code. ^b^ Density is population-based and estimated per 1000 inhabitants. Bolded numbers indicate statistical significance at the *p* < 0.05 level.

## Data Availability

De-identified data from this study are not available in a public archive. De-identified data from this study may be made available (as allowable according to institutional IRB standards) by emailing the corresponding author.

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
