# Peer review of "Examining Place-Based Neighborhood Factors in a Multisite Peer-Led Healthy Lifestyle Effectiveness Trial for People with Serious Mental Illness"

_ijerph, 2023, doi:10.3390/ijerph20095679_

Round 1
Reviewer 1 Report
This study, which analyzes the residential characteristics of PGLB participants in New York and Philadelphia, presents results that differ from the authors' original assumptions may be, but the reviewer believe that the paper is of high academic value and deserves to be accepted. In addition, the following points are questionable and should be considered.
(1) In the χ2 test, some of the cells have a frequency of 5 or less. In this case, the χ2 test may not be an accurate test, and other methods such as Fisher's exact probability test should be considered.
2) Although it may be an oversight by the reviewer, the t-test does not seem to have been used.
Author Response
- This study, which analyzes the residential characteristics of PGLB participants in New York and Philadelphia, presents results that differ from the authors' original assumptions may be. Still, the reviewer believes that the paper is of high academic value and deserves to be accepted. In addition, the following points are questionable and should be considered.
Response: We thank the reviewer for their thoughtful and helpful feedback to strengthen our work.
- In the χ2 test, some cells have a frequency of 5 or less. In this case, the χ2 test may not be an accurate test, and other methods, such as Fisher's exact probability test, should be considered.
Response: We thank the reviewer for this suggestion. We agree that Fisher’s exact test is better suited for many of our analyses. As such, we have replaced the prior Chi-Squared tests for Tables 1-4 with Fisher’s exact tests. Importantly, all findings remained consistent in direction and strength, with our prior results based on Chi-Squared tests. Only two associations of several tested had different results in terms of statistical significance (with consistent direction and strength to prior results): a) in Philadelphia, the spatial distribution of transit stop density is no longer found to be significantly different between study neighborhoods and those of the rest of the city (p=0.074 based on Fisher’s exact test; p<0.05 with Chi-squared test); b) in Philadelphia, the spatial distribution of hospitals was now found to be significantly different when contrasting PGLB study neighborhoods from those of the rest of the city (p=0.024 per Fisher’s exact test; p>0.05 per Chi-squared test). The text has been updated to reflect these findings and the method used. Further, we decided to keep the use of the Chi-squared test for the analyses shown in Tables 5 and 6, given that the cell sizes made it feasible and preferable to opt for this test.
- Although it may be an oversight by the reviewer, the t-test does not seem to have been used.
Response: Thank you. The mention of t-tests was a mistake. We have removed the text in the methods mentioning t-tests.
Reviewer 2 Report
This manuscript analyzes a meaningful issue that is particularly detrimental to the mitigation of SMI disadvantage in the United States, and I recommend it for publication, after refining the following issues
1. it is difficult to define overweight and obesity as the same criteria, in fact, more obesity studies consider BMI ≥ 24 means overweight, BMI ≥ 28 means obesity, and BMI ≥ 30 means severe obesity. i suggest caution to consider overweigt and obesity as the same. In line127.
2. I suggest adding the conceptualization operation of the variable after line 269, the type of the variable, i.e. whether it is continuous or binary, or the category.
3. The use of chi-square tests to indicate between-group heterogeneity only preliminary reveals a descriptive relationship between variables, and the addition of regression models is recommended. If the authors reject it, this preliminary relationship should be made explicit in the “research limitations”part.
4. I strongly recommend presenting your discussion in points, in the "Discussion" section of line 406 of the manuscript, where you can present your ideas one by one and compare them with prior experience.
5. The manuscript examines a meaningful topic and I suggest adding a policy recommendations section as your findings are key to alleviating the SMI population in the US. Please add policy recommendations under 570.
6. Similarly, please add your expectations for future research under line 570.
Author Response
We thank you for the opportunity to submit a revised version of our manuscript titled “Examining place-based neighborhood factors in a multisite peer-led healthy lifestyle effectiveness trial for people with serious mental illness” for consideration for publication in your journal.
We thank the reviewers for their thoughtful comments and suggestions, which we believe have strengthened our work.
We have considered all the suggested changes by reviewers, as well as by the Assistant Editor. Our point by point response to these comments is as follows:
- This manuscript analyzes a meaningful issue that is particularly detrimental to the mitigation of SMI disadvantage in the United States, and I recommend it for publication, after refining the following issues.
Response: We thank the reviewer for their thoughtful and helpful feedback to strengthen our work.
- It is difficult to define overweight and obesity as the same criteria, in fact, more obesity studies consider BMI ≥ 24 means overweight, BMI ≥ 28 means obesity, and BMI ≥ 30 means severe obesity. I suggest caution to consider overweight and obesity as the same. In line127.
Response: Thank you for this comment. We agree with the reviewer that the traditional BMI-based cut-points for overweight and obesity are not always accurate or ideal. However, because our study was a secondary analysis of previously collected data via the parent study (PGLB trial), and this is the categorization that was used in that study for eligibility purposes, this is how we are defining it. This said, in light of the reviewer’s valid comment, we have added clarification on this matter both in the description of the parent study (lines 127-129) and in the limitations paragraph within the discussion section (lines 552-555).
- I suggest adding the conceptualization operation of the variable after line 269, the type of the variable, i.e. whether it is continuous or binary, or the category.Response: Thank you for this suggestion. We have added a note in the text indicating that detailed information of variable operationalization is available in Appendix Table 1. Lines 202-206.
- The use of chi-square tests to indicate between-group heterogeneity only preliminary reveals a descriptive relationship between variables, and the addition of regression models is recommended. If the authors reject it, this preliminary relationship should be made explicit in the “research limitations” part.
Response: Thank you for this important comment. While we agree with the reviewer that adjusted regression models would be preferable, in particular for the analyses presented in tables 4-6, unfortunately, our sample of zip codes for study participants, based on the sampling frame of the original parent study, limited our ability to perform this type of analysis. Simply stated, we did not have sufficient statistical power to conduct more sophisticated analyses. We have now added text to our limitation section to explicitly acknowledge this. This is now mentioned in the Limitations paragraph of the discussion in lines 552-555.
- I strongly recommend presenting your discussion in points, in the "Discussion" section of line 406 of the manuscript, where you can present your ideas one by one and compare them with prior experience.
Response: Thank you for this helpful suggestion. We have now added sub-headings in the discussion section to highlight the different topics of contrast to previous literature throughout the discussion section.
- The manuscript examines a meaningful topic and I suggest adding a policy recommendations section as your findings are key to alleviating the SMI population in the US. Please add policy recommendations under 570.
Response: Thank you for this suggestion. This has been added to the conclusions paragraph. Lines 618-631.
- Similarly, please add your expectations for future research under line 570.
Response: thank you for this suggestion. This has been added to the closing paragraph of the discussion section. Lines 587-599.
Reviewer 3 Report
This paper presents an interesting theme on the influences of the place-based neighborhood factors to the health behaviors of the people with serious mental illness. Thought there is a limitation in the methodology as the authors admitted, there are some important outcomes that can be used to build healthy and equal cities for ethnic minorities. Hence, I would like to recommend this paper if the authors are able to address the following questions:
1. The structure of the paper looks good, but an overall framework is still missing. The paper can be better structured if the authors could complement a research framework in the second section.
2. It would be better if the authors could list the place-based factors and their measurement by a table. The present description makes readers more confused and unclear about how the authors assess these factors.
3. There are many places that need more references to support their ideas, i.e., in line 168-169, it is stated “variables were selected based on the types of neighborhood-level features that have been found to be associated with health behaviors in general population studies.” It is necessary to specialize what health behaviors and cite the references. Similarly, in line 180, it speaks “on the other hand, many intervention studies include sites distributed across multiple cities.”, but no reference was given.
4. I understand the paper is within the field health and behavior science, but since the topic is highly related to urban environment and design, it might be necessary to include several publications from urban science. There is also a need to provide some suggestions in the discussion on that how this study can help planners and designers to develop healthy and equal cities or communities for SMI people.
Author Response
We thank you for the opportunity to submit a revised version of our manuscript titled “Examining place-based neighborhood factors in a multisite peer-led healthy lifestyle effectiveness trial for people with serious mental illness” for consideration for publication in your journal.
We thank the reviewers for their thoughtful comments and suggestions, which we believe have strengthened our work.
We have considered all the suggested changes by reviewers, as well as by the Assistant Editor. Our point by point response to these comments is as follows:
- This paper presents an interesting theme on the influences of the place-based neighborhood factors to the health behaviors of the people with seriousmental illness. Thought there is a limitation in the methodology as the authors admitted, there are some important outcomes that can be used to build healthy and equal cities for ethnic minorities. Hence, I would like to recommend this paper if the authors are able to address the following questions below.
Response: We thank the reviewer for their thoughtful and helpful feedback to improve our work.
- The structure of the paper looks good, but an overall framework is still missing. The paper can be better structured if the authors could complement a research framework in the second section.
Response: Thank you for this suggestion. We have added text to explicitly acknowledge that our work is grounded on the socio-ecological model as our guiding theoretical framework (lines 54-77).
- It would be better if the authors could list the place-based factors and their measurement by a table. The present description makes readers more confused and unclear about how the authors assess these factors.
Response: Thank you for this suggestion. We have added text in the methods section indicating that a more detailed description of geospatial variables, including their operationalization and data sources, is included in Online Appendix Table 1.
- There are many places that need more references to support their ideas, i.e., in line 168-169, it is stated “variables were selected based on the types of neighborhood-level features that have been found to be associated with health behaviors in general population studies.” It is necessary to specialize what health behaviors and cite the references. Similarly, in line 180, it speaks “on the other hand, many intervention studies include sites distributed across multiple cities.”, but no reference was given.
Response: Thank you for this suggestion. Additional references have been added to support these and other statements throughout the manuscript.
- I understand the paper is within the field health and behavior science, but since the topic is highly related to urban environment and design, it might be necessary to include several publications from urban science. There is also a need to provide some suggestions in the discussion on that how this study can help planners and designers to develop healthy and equal cities or communities for SMI people.
Response: Thank you very much, this is an excellent suggestion. We have added both new references and language/ideas from the fields of urban design and planning to address this point in the Conclusions section. Lines 618-631.